# Coral reef restoration efforts in Latin American countries and territories

**Elisa Bayraktarov**[1]*, **Anastazia T. Banaszak**[2], **Phanor Montoya Maya**[3], **Joanie Kleypas**[4], **Jesús E. Arias-González**[5], **Macarena Blanco**[6], **Johanna Calle-Triviño**[5,6,7], **Nufar Charuvi**[8], **Camilo Cortés-Useche**[5,7], **Victor Galván**[9], **Miguel A. García Salgado**[10], **Mariana Gnecco**[3], **Sergio D. Guendulain-García**[2], **Edwin A. Hernández Delgado**[11,12,13], **José A. Marín Moraga**[4], **María Fernanda Maya**[3], **Sandra Mendoza Quiroz**[2,14], **Samantha Mercado Cervantes**[9], **Megan Morikawa**[6], **Gabriela Nava**[10], **Valeria Pizarro**[15], **Rita I. Sellares-Blasco**[7], **Samuel E. Suleimán Ramos**[11], **Tatiana Villalobos Cubero**[4], **María F. Villalpando**[7], **Sarah Frías-Torres**[16,17]

**1** Center for Biodiversity and Conservation Science, The University of Queensland, Brisbane, Queensland, Australia, **2** Investigación Integral para la Conservación de Arrecifes, Universidad Nacional Autónoma de México (UNAM), Puerto Morelos, Quintana Roo, Mexico, **3** Corales de Paz, Cali, Colombia, **4** Raising Coral Costa Rica/Centro de Investigación en Ciencias del Mar y Limnología (CIMAR), San Pedro, Costa Rica, **5** Departamento de Recursos del Mar, Centro de Investigación y de Estudios Avanzados del Instituto Politécnico Nacional, Mérida, Yucatán, Mexico, **6** Iberostar Hotels & Resorts, Wave of Change, Dominican Republic, Mexico, **7** Fundación Dominicana de Estudios Marinos (FUNDEMAR), Santo Domingo, Dominican Republic, **8** Fundación Calipso, Taganga, Magdalena, Colombia, **9** Fundación Grupo Puntacana (FGPC), Punta Cana, La Altagracia Province, Dominican Republic, **10** Oceanus A.C., Chetumal, Quintana Roo, Mexico, **11** Sociedad Ambiente Marino (SAM), San Juan, Puerto Rico, **12** Department of Environmental Sciences, University of Puerto Rico, San Juan, Puerto Rico, **13** Center for Applied Tropical Ecology and Conservation, University of Puerto Rico, San Juan, Puerto Rico, **14** SECORE International, Inc., Hilliard, Ohio, United States of America, **15** Fundación para la Investigación y Conservación Biológica Marina ECOMARES, Cali, Valle del Cauca, Colombia, **16** Vulcan, Seattle, Washington, United States of America, **17** Smithsonian Institution, Washington, D. C., United States of America

* e.bayraktarov@uq.edu.au

**Data Availability Statement:** The data is provided within the submitted supplementary information.

**Funding:** The funder provided support in the form of salaries for authors [JC-T, MB, MM and SMQ],

## Abstract

Coral reefs worldwide are degrading due to climate change, overfishing, pollution, coastal development, coral bleaching, and diseases. In areas where the natural recovery of an ecosystem is negligible or protection through management interventions insufficient, active restoration becomes critical. The Reef Futures symposium in 2018 brought together over 400 reef restoration experts, businesses, and civil organizations, and galvanized them to save coral reefs through restoration or identify alternative solutions. The symposium highlighted that solutions and discoveries from long-term and ongoing coral reef restoration projects in Spanish-speaking countries in the Caribbean and Eastern Tropical Pacific were not well known internationally. Therefore, a meeting of scientists and practitioners working in these locations was held to compile the data on the extent of coral reef restoration efforts, advances and challenges. Here, we present unpublished data from 12 coral reef restoration case studies from five Latin American countries, describe their motivations and techniques used, and provide estimates on total annual project cost per unit area of reef intervened, spatial extent as well as project duration. We found that most projects used direct transplantation, the coral gardening method, micro-fragmentation or larval propagation, and aimed to optimize or scale-up restoration approaches (51%) or provide alternative, sustainable

but did not have any additional role in the study design, data collection and analysis, decision to publish, or preparation of the manuscript. The specific roles of these authors are articulated in the 'author contributions' section.

**Competing interests:** On behalf of my co-authors, I confirm that the commercial affiliation of some of the authors with SECORE International, Inc. and Iberostar Hotels & Resorts does not alter our adherence to all PLOS ONE policies on sharing data and materials. This does not alter our adherence to PLOS ONE policies on sharing data and materials.

livelihood opportunities (15%) followed by promoting coral reef conservation stewardship and re-establishing a self-sustaining, functioning reef ecosystems (both 13%). Reasons for restoring coral reefs were mainly biotic and experimental (both 42%), followed by idealistic and pragmatic motivations (both 8%). The median annual total cost from all projects was $93,000 USD (range: $10,000 USD—$331,802 USD) (2018 dollars) and intervened a median spatial area of 1 ha (range: 0.06 ha—8.39 ha). The median project duration was 3 years; however, projects have lasted up to 17 years. Project feasibility was high with a median of 0.7 (range: 0.5–0.8). This study closes the knowledge gap between academia and practitioners and overcomes the language barrier by providing the first comprehensive compilation of data from ongoing coral reef restoration efforts in Latin America.

## Introduction

Active restoration is defined as the process of assisting the recovery of an ecosystem that has been degraded, damaged, or destroyed [1]. It may be increasingly necessary on coral reefs, once it has been determined that the natural recovery of corals is hindered [2]. In comparison, rehabilitation is typically described as the replacement of structural or functional characteristics of an ecosystem that have been diminished or lost [3]. As for any conservation intervention, setting clear goals and defining indicators to measure progress towards these goals is of pivotal role in judging success [4]. The goal of any restoration action is to eventually establish self-sustaining, sexually reproducing populations with enough genetic variation enabling them to adapt to a changing environment [5–7].

Coral reef restoration may play a particularly important role where coral species are threatened with extinction. The Caribbean Elkhorn coral, *Acropora palmata*, and Staghorn coral, *A. cervicornis*, were once widely distributed and among the major reef-building species in the region [8]. Both species are now listed as Critically Endangered on the International Union for Conservation of Nature (IUCN) Red List [9] as a result of major losses in cover of both species throughout the Caribbean since the 1970s [10].

The lack of natural recovery of Caribbean coral reefs [11] has spurred the need for active management programs to assist in their recovery [12, 13]. Management actions include effective spatial planning, enforcement, no take zones, treatment of sewage and protection of adjoining ecosystems such as mangroves [12, 14–16]. Resilience-based management of coral reefs [17] may stimulate coral recovery, especially if applied in conjunction with active restoration [13, 18]. The rationale being that seeding corals onto reefs where larval supply or post-settlement survival have been inadequate, will only be successful if the conditions are suitable for supporting their survival and growth.

Several techniques are used for the restoration of coral reefs. The most common techniques are based on asexual methods such as direct transplantation, coral gardening, and micro-fragmentation [19]. An alternative technique, larval propagation, is based on the collection of gametes and the consequent culturing of embryos and larvae, after which the coral spat are either grown in *ex situ* aquaria to larger-sized colonies or are outplanted onto degraded reefs at approximately one month old [20]. While the techniques used to restore coral reefs are reviewed elsewhere (e.g. [19, 21–23]), here we focus on direct transplantation, coral gardening, micro-fragmentation, and larval propagation as the techniques most-commonly employed by the case studies in the study area. One of the oldest techniques used in coral reef restoration is direct transplantation of corals [24], which involves the harvesting of coral colonies from a

donor site and their immediate transplantation to a restoration site or re-attaching colonies that have been dislodged by a ship grounding, storm or hurricane [25]. The coral gardening approach was developed to scale-up restoration while reducing the stress on donor colonies. Fragments of corals are harvested from donor colonies, grown in nurseries to a threshold size [18] before being transplanted onto a degraded reef [26, 27]. Nurseries can be ocean-based (*in situ*) or land-based (*ex situ*). *In situ* nurseries are typically located in sheltered environments where conditions are favourable for coral growth and safe from predation, storm surges, and wave energy, and are regularly maintained and cleaned by physical removal of algal growth [28]. However, strategic siting of ocean nurseries can promote the recruitment of fish assemblages that remove biofouling through grazing, thus may significantly reduce person-hours spent in nursery cleaning [29]. *In situ* nurseries can have many shapes and sizes. For example, they can consist of floating mid-water structures built using ropes, mesh or cages [29–32], structures placed on concrete, tables or frames [33], PVC 'trees' [34], PVC grids or dead coral bommies [35]. *Ex situ* nurseries typically use flow-through large aquaria or raceways, and require continuous access to electricity, water quality monitoring, and control of temperature and light availability [36]. Micro-fragmentation is an approach especially useful for slow-growing massive corals. This technique involves the fragmentation of parts of a massive coral donor to yield multiple ~1 cm$^2$ fragments. The fragments are placed close to each other on either artificial substrates or on the surface of dead coral colonies. The micro-fragments, as they recognize neighbouring fragments as kin, grow towards each other and fuse [37]. Ideally, they are outplanted to the degraded reef at a size of ~6 cm$^2$ [37, 38]. Larval propagation involves the breeding of corals from eggs and sperm. Studies describing this technique typically report the use of raceways with seawater flow-through systems where coral spawn is collected from the wild, fertilization is assisted, embryos are cultured to larvae, which are settled onto substrates and then transported and seeded onto a degraded coral reef [39–42]. This process has also been referred to as larval enhancement, sexual propagation, sexual coral cultivation or larval reseeding [21]. As an emerging larval propagation technique, larval restoration concentrates coral larvae over enhancement plots on the degraded reef to facilitate coral larvae settlement directly to the substrate, without the need for laboratory facilities [43]. The first attempts to use larval seeding on the reef have been developed only recently (in 2002, [44]) and it is still a matter of active debate whether direct seeding of mass-cultured coral larvae is an effective option for reef rehabilitation [43, 45]. The main advantages of the larval propagation techniques are that they increase the genetic diversity among restored coral populations thus enabling increased rates of adaptation and improved resilience in the context of climate change [46], and they have the potential to be used over large scales while reducing the cost [39]. Also, they do not cause damage to the parent colonies when gametes are collected *in situ* with nets or from spawn slicks without removing the gravid colonies from their location.

While efforts in the USA, Australia or places where European scientists conduct their research are well described in the published literature and disseminated at conferences, there is a paucity of documentation on coral reef restoration projects carried out by practitioners in the Caribbean and Eastern Tropical Pacific. Reasons for this lack of exchange may be the language barrier, lack of interest in knowledge transfer between higher and lower income countries or cultural differences as well as lack of funding. In 2018, the Reef Futures symposium was held in the Florida Keys, USA and attended by over 400 delegates. The aim of this international meeting was to 'bring together experts from around the world to share the latest science and techniques for coral reef restoration while kicking off a global effort to dramatically scale-up the impact and reach of restoration as a major tool for coral reef conservation and management'. The conference was organized by the Coral Restoration Consortium, which is a community comprised of scientists, managers, coral restoration practitioners, and educators

dedicated to enabling coral reef ecosystems to survive the 21st century and beyond' [47]. Within the Reef Futures conference, we convened a meeting of scientists and practitioners involved in active coral reef restoration in the Latin- and Centro-American Caribbean as well as the Eastern Tropical Pacific to fill the knowledge gap between academia and practitioners in the region and overcome the language barriers in coral reef restoration. Here, we showcase the advances and share the lessons learned from 12 restoration case studies from the Caribbean and Eastern Tropical Pacific. We provide a comprehensive compilation of unpublished data from coral reef restoration efforts where we outline the techniques that were employed, the motivations and objectives of each project, total project cost per unit area per year, spatial extent of intervention, project duration, and the indicators of success measured. This work provides the most complete data set on total project cost, feasibility and success indicators of coral reef restoration from practical cases that may guide decisions required to establish new restoration projects in the future.

## Approach

### Data collection

The co-authors of this work contributed data and descriptions of their restoration projects which constitute the case studies used here. The coral reef restoration projects were carried out in Latin American countries and territories in the Caribbean and Eastern Tropical Pacific (Fig 1). The data obtained included estimates on total annual project cost, spatial extent of area intervened, project duration, an estimate on the project reaching specific objectives within a fixed period of time and the biotic, socio-economic and legislative indicators of success used to track restoration progress. The motivations for each restoration project were adopted from [19, 48, 49] and classified as biotic, experimental, idealistic, legislative, and pragmatic (Table 1).

The objectives of coral reef restoration projects can be highly diverse and dependent on the specific project as well as its location. In this study, the restoration practitioners were asked to provide the objectives for their restoration projects, which were specific, measurable,

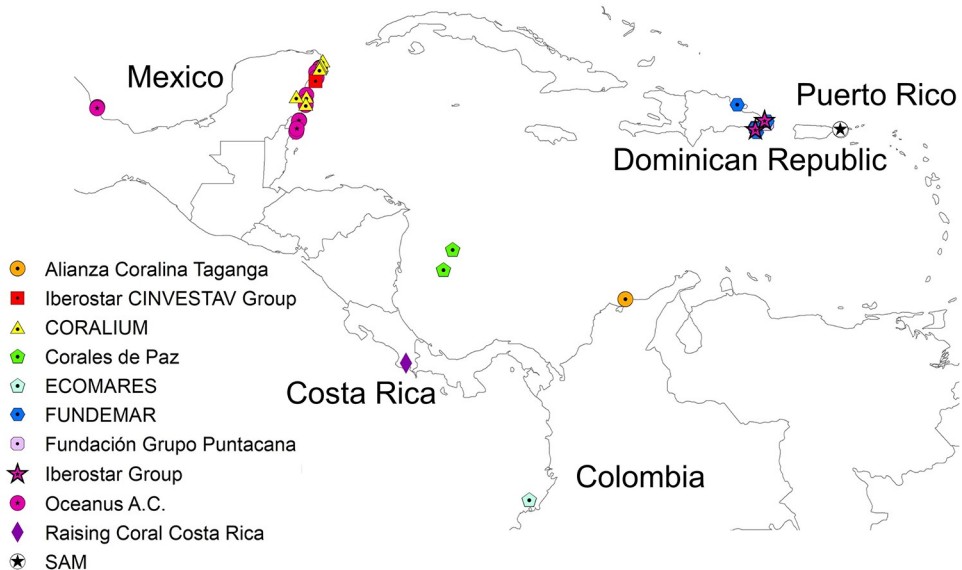

**Fig 1. Map of coral reef restoration projects in Spanish-speaking Latin American countries and territories.**

**Table 1. Five motivation categories for carrying out coral reef restoration projects and examples.**

| Motivation category | Examples |
| --- | --- |
| Biotic | Biodiversity enhancement (e.g., native species, habitat creation, ecosystem connectivity, ecological resilience) |
| Experimental | Improve restoration approaches, technologies, and methods. Answer ecologically-based research questions |
| Idealistic | Cultural reasons (e.g., recreation, tourism, medicinal/ceremonial substances, spiritual importance, aesthetic value) |
| | Social reasons (e.g., community involvement, job creation, nature education, environmental outreach) |
| | Political reasons (e.g., raising environmental profile) |
| Legislative | Restoration after environmental impact (e.g., ship-grounding, mining, oil spill, hurricane damage) |
| | Biodiversity offset (e.g., threatened species, threatened ecological communities) |
| Pragmatic | Enhance ecosystem services (e.g., fisheries production) |
| | Enhance ecosystem services (e.g., water quality improvement, pollution prevention) |
| | Enhance ecosystem services (e.g., coastal protection, erosion control, bank stabilisation) |
| | Enhance ecosystem services (e.g., carbon sequestration, carbon offsets) |

achievable, repeatable and time-bound (SMART; [5]). We modified the six primary objectives observed by Hein et al. [50] into the following categories: 1) enhance ecosystem services for the future; 2) optimize/scale-up restoration approaches; 3) promote coral reef conservation stewardship; 4) provide alternative, sustainable livelihood opportunities; 5) reduce coral population declines and ecosystem degradation; and 6) re-establish a self-sustaining, functioning reef ecosystem.

The total estimated project cost includes both capital and operating costs. Capital costs are those used for planning, land acquisition, construction, and financing [51]. These may also include costs for laboratory/infrastructure, boats and dive equipment. Operating costs are those used for maintenance, monitoring, equipment repair and replacement [51] and may include salaries, housing for scientific/implementation teams, air for SCUBA tanks, gasoline for boat engines, and replacement of computers. Coral reef practitioners were asked to estimate the total cost for restoration interventions based on the guidelines for standardised reporting of costs for management interventions for biodiversity conservation [52] and are provided as United States Dollars (USD) per hectare of coral reef intervened per year in 2018 USD.

The project spatial extent is the coral reef area intervened by the restoration project and is reported in hectares. Spatial extent is not provided for each project since not all restoration case studies have an objective to increase the area of restored habitat. For instance, some projects are aimed at developing new restoration techniques, using coral nurseries as a tool to stimulate public awareness and engagement, for educational purposes, or as a tourist attraction.

The project duration is the time during which the restoration project has existed until the present, or the time during which the restoration cost was budgeted for and is provided in years. All projects described here are ongoing and active throughout 2019.

The feasibility is the likelihood that each specific project objective can be reached successfully with the interventions at hand and within the outlined project duration. It is ideally measured as the likelihood of success in returning the ecosystem function and resilience of an ecosystem through restoration [53]. This overall restoration project feasibility is rarely reported in the published literature because a standardised method to measure restoration

success is largely missing [51]. Here, restoration practitioners estimated the feasibility of the restoration interventions they employed to achieve their specific project objectives. Feasibility is given as a ratio between 0 and 1 and can be interpreted as the likelihood of success to reach a specific objective within the duration of the restoration project. Practitioners provided a minimum, maximum and the best guess for the project feasibility.

Each case study defined their indicators of success which were monitored throughout the lifetime of the project. These were categories into the groups: biotic, socio-economic and legislative success indicators (Table 2).

## Results

Data from a total of 12 coral reef restoration projects carried out by practitioners in the Spanish-speaking Caribbean and Eastern Tropical Pacific were compiled and are summarised in Table 2. The supplementary material contains more detailed information about each restoration case study. Information was gathered from Colombia (Alianza Coralina Taganga, Corales de Paz, and ECOMARES), Costa Rica (Raising Coral Costa Rica), the Dominican Republic (FUNDEMAR, the Iberostar Group, and Fundación Grupo Puntacana), Mexico (Oceanus A. C., CORALIUM at Universidad Nacional Autónoma de México, and the Iberostar & CINVESTAV Group), and Puerto Rico (Sociedad Ambiente Marino) (Fig 1). Note that the Fundación Grupo Puntacana has two restoration programs of which one is focused on coral gardening (Program 1) and one is directed towards micro-fragmentation (Program 2). These were treated as independent projects for analytical purposes. The restoration projects use techniques that include direct transplantation (one project, 9%), coral gardening (7 projects, 64%), micro-fragmentation (5 projects, 45%), and larval propagation (2 projects, 18%) (Fig 2). Some projects also apply a combination of techniques e.g. direct transplantation, coral gardening and micro-fragmentation or coral gardening and micro-fragmentation as well as coral gardening and larval propagation (S1 Table in S1 File).

The primary motivations to carry out the coral reef restoration projects are biotic and experimental both with 41.7%, followed by idealistic and pragmatic reasons (both 8.3%). Biotic (36.3%) and experimental (27.3%) reasons were important secondary motivations, followed by legislative reasons (18.2%), and pragmatic/idealistic motivations (both 9.1%) (Fig 3). All except for one of the projects reported secondary motivations. The tertiary motivations reported by 5 of the 12 projects were mainly pragmatic (80.0%) and idealistic (20.0%).

Most projects have specific objectives to optimize/scale-up restoration approaches (51.1%), followed by providing alternative, sustainable livelihood opportunities (14.9%), and then in equal parts to promote coral reef conservation stewardship and re-establish a self-sustaining, functioning reef ecosystem (12.8%). The objectives to enhance ecosystem services for the future and the reduction of population decline and ecosystem degradation accounted for only 4.2% each of the specific project objectives.

The median total cost from all projects per year is $93,000 USD (± $32,731 SE) ranging between $10,000 USD and $331,802 USD. The median spatial extent of coral reef restoration intervention is 1.0 ha (± 1.3 ha SE) ranging between 0.06 ha and 8.39 ha. Project duration was as short as 1 year and could be as long as 17 years with the median project duration of 3 years (± 1.5 years SE) to reach the project targets. Projects reported a median feasibility of 0.7 (± 0.03 SE) ranging from 0.5 to 0.8 (Table 3).

## Discussion

Here we present the first comprehensive assessment of coral reef restoration projects in Spanish-speaking countries and territories of the Caribbean and Eastern Tropical Pacific (ETP),

**Table 2. Summary of the 12 restoration projects in the Caribbean and Eastern Tropical Pacific.** Cost values are given in 2018 USD.

| Country, Location, Organization | Technique employed (type of nursery) | Targeted coral species | Strategy for outplanting and monitoring | Spatial extent of project | Estimated project budget and funding bodies/ partners | Estimated project feasibility | Success indicators |
|---|---|---|---|---|---|---|---|
| | | | | **Implemented and in progress as of 2019** | | | |
| Colombia, Taganga, Caribbean Sea, Alianza Coralina Taganga | Coral gardening with one mid-water ocean nursery.Micro-fragmentation. | Mcav, Ppor Millepora spp. | **Outplanting:** drill holes in natural substrate, insert cement cookies using plastic nails. **Monitoring:** at least once per month for self-attachment to natural substrate. | Pending permits. | **Budget estimate:** $500,000 USD for 2 years. **Partners:** local stakeholders (40% of budget). | best guess = 0.5 minimum = 0.2 maximum = 0.9 | No information available yet. |
| Colombia, San Andres and Providencia Islands, Caribbean Sea, Corales de Paz | Coral gardening with midwater rope nurseries. Micro-fragmentation | Acer, Apal, Mcom, Mdec, Past, Peli, Ppor | **Outplanting:** use mix of marine cement and colloid adjuvant to improve fluidity and reduce runoff. **Outplanting density:** 5,000 corals/ha of degraded reef. **Monitoring:** ecological and structural surveys complemented with 3D imagery—photomosaics, carried out before, 3 months after, and 12 months after outplanting. | 6 hectares by year 4, distributed as 3 hectares at each of two islands. | **Budget estimate:** $900,000 USD with $37,500 USD ha$^{-1}$ yr$^{-1}$ annually. **Partners:** Agriculture and Fisheries Secretariat, Government of San Andrés Archipelago, Providencia and Santa Catalina, CORALINA, Conservation International Colombia, and Corales de Paz, MasBosques and BanCO2. | best guess = 0.6 minimum = 0.5 maximum = 0.9 | **Biotic:** This project raised 13,468 nursery-grown Acer, Apal, Ppor and Mdec in 8 mid-water floating rope nurseries (Oct 2017 to Dec 2018). Coral fragment survival was 79.41% (Jul 2019). By Oct 2019, 5,504 nursery-grown corals were outplanted. Outplant area was over 1,500 m², whereas ecological footprint (i.e. the maximum areal extent of outplant plots) was over 9,000 m². Increase in live coral cover of 23% in 2018 and 41% by the end of 2019. **Socio-economic:** 28 fishermen and over 25 recreational divers have been certified in large-scale coral reef restoration. |
| Costa Rica, Golfo Dulce, Eastern Tropical Pacific, Raising Coral Costa Rica/ Centro de Investigación en Ciencias del Mar y Limnología | Coral gardening with ocean tree and rope nurseries. Micro-fragmentation. | Pewe, Pfro, Pgig, Plob Pocillopora sp. Psammocora sp. | **Outplanting:** cable ties attached to large nails onto substrate. Planned: outplant corals still connected by ropes. Corals on cement plugs outplanted into holes drilled into the substrate. | 10 reef patches of 200–500 m² each within the next 3 years for a maximum of 0.5 ha. | **Budget estimate:** $120,000 USD (2.5 years). In 2018, the annual project cost was $35,000 USD. **Partners:** The project is mainly financed by private donations. | best guess = 0.8 minimum = 0.6 maximum = 0.9 | **Biotic:** 2,000 corals were propagated in 10 in situ coral structures over 2 years: 379 from three different species were successfully outplanted. Outplanted corals had a 78% survival over 2 years (Outplants: 174 Pocillopora sp. with 91% survival; 85 Porites spp. with 68% survival; 120 Pgig with 73% survival). **Socio-economic:** Project engaged with 2 local hotels and trained 7 local persons in coral maintenance of coral nurseries and outplanting over 2 years. |
| Dominican Republic, Bayahibe, Caribbean Sea, FUNDEMAR | Coral gardening with 8 rope and steel rod ocean nurseries. Larval propagation in situ (SECORE floating pools). Larval propagation ex situ in a wet laboratory. | Acer, Apal, Cnat, Dcyl Dlab, Oann, Ofav | **Outplanting:** attach corals to substrate with nails, cable ties and epoxy. Acer at 1 coral colony per m2, for 2,000 colonies each 20–30 cm in diameter. Corals from larval propagation settled on cement or ceramic substrates are seeded onto reef. 2,000 recruits seeded of 3–5 species. **Monitoring:** Fragment counts on all Acer transplant sites and monitoring of survival of sexual recruits through time. | At least 0.5 hectares of degraded coral reef per year. Completed: Coral outplanting at 12 restoration sites. | **Budget estimate:** $51,800 USD per year. **Partners:** Private and public national and international institutions and grants and alliances with other organizations carrying out coral reef restoration. | best guess = 0.7 minimum = 0.4 maximum = 0.9 | **Biotic:** From 2011–2017 this project grew 26,000 cm of tissue in 8 in situ coral nurseries, of which 1,446 cm were successfully outplanted in 6 sites. One year after outplanting, the transplants had a 71.6 ± 10.4%, with a range of 57.3–83.3%, survival. There have been more transplant events after 2017 but this information has not been processed. **Socio-economic:** Over 5 years, this project engaged with 7 local dive shops and trained more than 100 local fishermen, dive center personnel and volunteers to do maintenance on the coral nurseries and outplant corals. |
| Dominican Republic, Bayahibe, Caribbean Sea, The Iberostar Group | Coral gardening ex situ and in situ. | Aaga, Acer, Apal, Dlab, Oann, Ofav, Past, Ppor | Under development | Under development | **Budget estimate:** $100,000 USD over 10 months. In 2018, $40,000 USD were spent on construction (excluding salaries). **Partners:** Wave of Change movement, FUNDEMAR, University of California at Santa Barbara. | best guess = 0.5 minimum = 0.2 maximum = 0.8 | **Envisioned success indicators: Biotic:** 50 structures in in situ nursery, of which 31 structures belong to our gene bank. 100% survival rate in in situ nursery till March 2020. 10 species maintained in an ex situ genetic bank. Once outplanted, the following success indicators will be measured: survival rate, annual growth, recruitment rate, sexual maturity of outplanted colonies, abundance and richness of reef fish, functional diversity and evolutionary history **Socio-economic:** 3,296 guided tours to the Coral Lab were organised with an average of 470 visitors/ month between August 2019 to March 2020. |

*(Continued)*

**Table 2.** (Continued)

| Country, Location, Organization | Technique employed (type of nursery) | Targeted coral species | Strategy for outplanting and monitoring | Spatial extent of project | Estimated project budget and funding bodies/partners | Estimated project feasibility | Success indicators |
|---|---|---|---|---|---|---|---|
| Dominican Republic, Punta Cana, Caribbean Sea, Fundación Grupo Puntacana (Program 1) | Coral gardening using A-frames, tables and ropes. | *Aapa, Acer, Apal, Orbicella* spp. *Porites* spp. *Pseudodiploria* spp. | Outplanting: cable ties and galvanized nails. | Since 2011, a total of 9,425. *A. cervicornis colonies* (representing 5,635 linear meters of coral tissue; average. fragment size of 0.65m) have been transplanted over almost 0.44 ha of degraded reef. | Estimated budget: $93,000 USD in 2018, and $211,363 USD ha$^{-1}$ yr$^{-1}$ when extrapolated from the actual area intervened (0.44 ha.) The total estimated budget for the time interval 2019–2021 will be approximately $950,000 USD, thus equalling the total cost of $313,500 USD ha$^{-1}$ yr$^{-1}$. | best guess = 0.8 minimum = 0.5 maximum = 0.9 | Biotic: Since 2011, interns and local fishermen have established over 79 experimental transplant sites along 16km of coastline in the Punta Cana region in the eastern Dominican Republic. During this time, 9,425 coral colonies equivalent to 5,635 linear meters were returned back onto natural reefs. A survey of approximately 39% of the established transplant sites (2011–2018) was performed with over 1,188 fragments being identified for an estimated linear extension of 1,538 meters (due to the methodology used, this value is highly conservative). Mortality ranged from 0.0% to 65.3% (one plot within a site—high wave energy and high sedimentation). Average mortality for the sites surveyed was 17.3%. Additionally, hundreds to potentially thousands of coral fragments were outplanted based on donations from government agencies and civil society organisations. Socio-economic: In the last 6 years, more than 200,000 national and international students have been made aware of the foundation's coral restoration and conservation programs. Ten dive instructors from three Caribbean countries have been certified in teaching coral restoration techniques through the Coral First Aid Distinctive Specialty Course. Knowledge exchanges have taken place with 10 Caribbean Nations. Over the last 4 years, six fishermen have been moved into full-time coral restoration positions helping to prevent an estimated 8.8km/day/fisherman of parrotfish from being caught. These fishermen have transplanted 3,904 colonies (3,088 linear meters). In Nov 17–21, 2019, 28 administrators, practitioners, coral gardeners, tourists and volunteers from the Dominican Republic gathered in Bavaro to transplant ~1,660 coral fragments onto local reefs. To our knowledge, this was the first time that aerial mapping was used to guide outplanting efforts. |
| | | | | | Partners: Private donors, national and international grants and institutions such as Deutsche Gesellschaft für Internationale Zusammenarbeit (GIZ), The Nature Conservancy (TNC), Counterpart International (CPI), Caribbean Hotel and Tourism Association, Global Giving, and InterAmerican Development Bank (IDB). | | |
| Dominican Republic, Punta Cana, Caribbean Sea, Fundación Grupo Puntacana (Program 2) | Coral gardening. Micro-fragmentation. | *Mcav, Oann, Past, Pcli, Pfur, Pstr* | Outplanting: using established protocols for micro-fragments. | By the end of the third phase, an estimate of 5,000 micro-fragments will be outplanted annually covering up to 200 m$^2$ per year. | Estimated budget: $30,000 USD (2018). The project duration is three years and the total estimated budget is $850,000 USD (pending grant approvals). Partners: as above. | best guess = 0.6 minimum = 0.4 maximum = 0.9 | No information available yet. |
| Mexico, Chetumal, Caribbean Sea, Oceanus A. C. | Coral gardening. Fragments of opportunity. | *Acer, Apal, Apro, Agaricia* spp. *Diploria* spp. *Orbicella* spp. *Porites* spp. | Monitoring: carried out before and after transplantation to evaluate the survival and growth of restored corals. | Since 2014 to date, the estimate is 6.3 ha. | Estimated budget: $150,000 USD ha$^{-1}$ yr$^{-1}$ since 2014. | best guess = 0.8 minimum = 0.5 maximum = 0.9 | Biotic: 52,053 coral colonies were added to the Meso-american Reef (2014–2019) 85% survival of outplanted colonies was measured at end of 2019. Average coral cover ranging between 2 and 5% at all sites pre-restoration (2014) increased to an average of 8.4% in 2019. Increases were found from 8 to 14% in 2019. Increases were found from 6% to 11% at Rodman Sur in Puerto Morelos; from <2 to 11% at Mayakobá (Playa del Carmen) between 2014 and 2019; and from 0 to 6% at La Poza (Xcalak North) just in 2019. At the oldest restoration sites initiated in 2013 and maintained by the program, the average size of outplanted coral fragments, increased from 7–10 cm to 30 cm in diameter average in 2019. Some outplants have reached a diameter of up to 110 cm. About 30% of the transplants evaluated in 2019 at all sites had a size of 20 cm in diameter on average indicating that they have reached a reproductive size. Socio-economic: 11 local restoration teams have received training between 2014 and 2019 on restoration techniques, 6 of which continue to actively support the maintenance of the nurseries and participate in multiple outplanting and monitoring events throughout the year. Participants from the tourism industry have also added a restoration component within their program activities aimed at generating income. |
| | | | | | Partners: Comisión Nacional de Áreas Naturales Protegidas (CONANP), Summit Foundation, the Mesoamerican Reef Fund, Ciudad Mayakobá Group, with local partners such as Acuario de Veracruz, Fundación de Parques y Museos de Cozumel, hotels from Playa del Carmen (Mayakobá chain) and from Mahahual and Xcalak, the Xcalak community, and tourist services providers from Cozumel, Puerto Morelos and Veracruz. | | |

(*Continued*)

**Table 2.** (Continued)

| Country, Location, Organization | Technique employed (type of nursery) | Targeted coral species | Strategy for outplanting and monitoring | Spatial extent of project | Estimated project budget and funding bodies/ partners | Estimated project feasibility | Success indicators |
|---|---|---|---|---|---|---|---|
| Mexico, Mexican Caribbean, Caribbean Sea, CORALIUM, Universidad Nacional Autónoma de México | Larval propagation *ex situ*. Seeding of coral sexual recruits onto degraded reefs. Coral sperm cryopreservation. | Apal, Dlab, Oann, Ofav, Pstr | **Outplanting**: artificial substrates with settled coral larvae are outplanted two to four weeks post-settlement (one-polyp stage). The substrates are placed into natural gaps formed by the reef framework. **Monitoring**: monthly for six months and bi-monthly for next six months then twice per year. | Area of outplants of one polyp sized sexual recruits is 0.17 hectares. | **Estimated budget**: From 2014 and 2018, the budget is estimated at $15,000 USD per year and equals $100,000 USD ha⁻¹ yr⁻¹. <br> **Partners**: Comisión Nacional de Áreas Naturales Protegidas, Consejo Nacional de Ciencia y Tecnología, Comisión Nacional para el Conocimiento y Uso de la Biodiversidad, Alianza World Wildlife Fund–Fundación Carlos Slim, SECORE International, The Nature Conservancy and Experiencias XCARET. | best guess = 0.7 minimum = 0.6 maximum = 0.9 | **Biotic**: Over 5 years, this project has outplanted corals to nine reefs along the Mexican Caribbean. The corals have ranged in age from 2 weeks to 3 years. Corals that were older at outplanting have higher survival (100%) after 4 years and at least 50% are reproductively active as evidenced by the production of gametes. The youngest corals at outplant (2 weeks) have much lower survival rates (<0.1%) and the surviving colonies, at time of writing, range from 7 months to 6 years old. Sperm from four coral species have been cryobanked. **Socio-economic**: Over a 10-year period, a capacity-building program has been established, involving over 150 participants from 14 countries in hands-on workshops and graduate courses. |
| Puerto Rico, Culebra Island, Caribbean Sea, Sociedad Ambiente Marino | Coral nurseries (trees). Micro-fragmentation. Direct coral cuttings. | Acer, Apal, Apro, Dcyl, Efas, Maur, Oann, Ofav, Pdiv | **Outplanting**: portland marine cement mixed with lime to neutralize pH. Cable ties and masonry nails are also used in the case of Acer. Direct outplanting by wedging coral fragments to the bottom and other successful methods in the past. **Outplanting density**: *Acer*, four per m², other species, one colony per four m². | The project has intervened an area of ca. 6 ha. The projected spatial extent of reef rehabilitation by year 2023 in total will be 8.4 ha. | **Estimated budget**: $1,327,206 (2018 USD) from 2019 to 2023, resulting in $158,189 USD ha⁻¹ yr⁻¹. ($50.26 USD per coral colony). (including community-based in-kind support) for the period of 2019 to 2022 is $2,311,280 (2018 USD) resulting in a total annual expenditure of $275,480 (2018 USD) ha⁻¹ or a total estimated expense of $87.53 per coral colony). <br> **Partners**: Center for Applied Tropical Ecology and Conservation (CATEC), University of Puerto Rico–Río Piedras Campus, Puerto Rico Department of Natural and Environmental Resources (PRDNER), and NOAA Restoration Centers (NOAA-RC). | best guess = 0.7 minimum = 0.5 maximum = 0.9 | **Biotic**: Over 15 years (prior to hurricanes), this project grew 70,000 *Acer* coral transplants in 3 *in situ* coral nurseries, of which 60,000 were successfully outplanted. One year after outplanting, the transplants had 80 to 95% survival. However, in areas that were subsequently impacted by deforestation, urban construction and runoff survival declined to 40–60%. *Acer* coral cover at the plot scale in restored locations increased from <1 to 10% within one year and to 20 to 25% within 2 to 3 years. Coral density also increased from about 1 per 200 m³ to 4 per m². Over 3 years, this project used a combination of fragments at risk and coral cuttings to increase *Apal* density from 1 coral per 50 m² to about 1 per 4 m². Over one year, this project used micro-fragmentation technique to increase *Dcyl* density from 0 to 1 coral per 2 m² at one location. **Socio-economic**: Over 15 years, this project engaged with the Culebra Island Fishers Association, with 10 NGOs and other private institutions, one dive shop, and trained one full time fisherman, and over 200 volunteers to maintain coral nurseries and outplant corals. |

**Planned work**

| Country, Location, Organization | Technique employed (type of nursery) | Targeted coral species | Strategy for outplanting and monitoring | Spatial extent of project | Estimated project budget and funding bodies/ partners | Estimated project feasibility | Success indicators |
|---|---|---|---|---|---|---|---|
| Colombia, Gorgona National Natural Park, Eastern Tropical Pacific, ECOMARES | The project is presently gathering scientific information for future needs on coral restoration in the area. | Pcla, Pdam | **Outplanting**: Portland cement mixed with sand and freshwater. | Not available yet. | **Estimated budget**: $10,000 USD (2018). <br> **Partners**: Universidad del Valle, Universidad Javeriana de Cali, and Gorgona National Natural Park. | best guess = 0.7 minimum = 0.5 maximum = 0.9 | **Biotic**: 230 *Pdam* were outplanted directly into a coral community area (El Remanso). After 366 days survival rate was 67%, with the smallest fragments surviving the highest (76%) and the larger fragments surviving the lowest (56%). In a line nursery 152 *Pdam* fragments were reared for 134 days. Survival rates was 66.9%, having the highest survival, linear growth and weight in the larger fragments (>80%). |
| Mexico, Cozumel National Natural Park, Caribbean Sea, The Iberostar & CINVESTAV Group: | Coral nurseries. | Acer, Apal, Dlab Orbicella spp. Pseudodiploria spp. Siderastrea spp. | Under development. | Under development. | **Estimated budget**: To be determined. <br> **Partners**: Cozumel National Natural Park (PNAC), Comisión Nacional de Áreas Naturales Protegidas (CONANP), Consejo Nacional de Ciencia y Tecnología (CONACYT) and the Mexican Secretariat of Environment and Natural Resources (SEMARNAT). | To be determined | **Envisioned success indicators: Biotic**: At future outplanting sites, survival rate, annual growth, recruitment rate, sexual maturity of outplanted colonies, abundance, richness of reef fish, abundance of *Diadema* spp., functional diversity and evolutionary history will be measured. **Socio-economic**: Theoretical and practical training in biology, ecology and identification of local coral species as well as restoration of threatened species, will be given to 50 people from the local community of Cozumel, including tourism service providers, boat captains, university students, diving instructors, government authorities, staff from different NGOs and staff from Iberostar. |

Note that here, the Fundación Grupo Puntacana program is represented within two separate projects. More detailed information can be found in the supplementary material. Abbreviations: Fundación Dominicana de Estudios Marinos, Inc. (FUNDEMAR), Fundación Grupo Puntacana (FGPC), SECORE International (SECORE) and Sociedad Ambiente Marino (SAM). Species abbreviations: Aaga *Agaricia agaricites*, Acer *Acropora cervicornis*, Apal *Acropora palmata*, Apro *Acropora prolifera*, Cnat *Colpophyllia natans*, Dcyl *Dendrogyra cylindrus*, Dlab *Diploria labyrinthiformis*, Efas *Eusmilia fastigiata*, Maur *Madracis aurentenra*, Mcav *Montastraea cavernosa*, Mcom *Millepora complanata*, Mdec *Madracis decactis*, Oann *Orbicella annularis*, Ofav *Orbicella faveolata*, Past *Porites astreoides*, Pcla *Pseudodiploria clivosa*, Pdam *Pocillopora damicornis*, Pdiv *Porites divaricata*, Peve *Porites evermanni*, Pfro *Pavona frondifera*, Pfur *Porites furcata*, Pgig *Pavona gigantea*, Plob *Porites lobata*, Ppor *Porites*, Pstr *Pseudodiploria strigosa*.

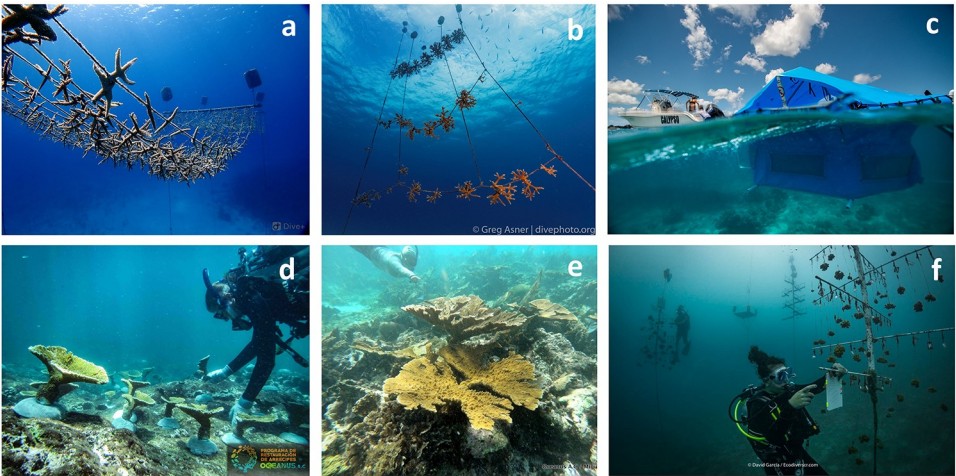

**Fig 2. Types of nurseries described in the text.** a) Floating rope nurseries used in San Andrés and Providencia islands for large-scale coral gardening (Photo: Corales de Paz); b) rope nurseries by FUNDEMAR in Dominican Republic (Photo: Greg Asner); c) FUNDEMAR's floating *in situ* coral larvae rearing tank (Photo: Paul Selvaggio); d) Oceanus A. C. diver outplants nursery grown corals in Veracruz, Mexico (Photo: Oceanus A.C.); e) outplanted *Acropora palmata* coral in Puerto Morelos, Mexico (Photo: Oceanus A.C.); Raising Coral Costa Rica's tree nurseries in Costa Rica (Photo: David Garcia).

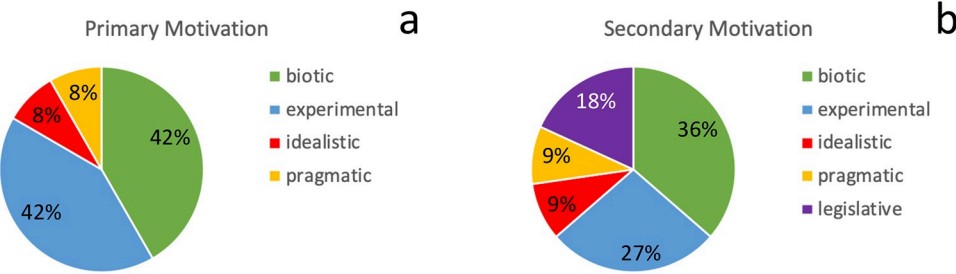

**Fig 3. Percentage of motivation categories (biotic, experimental, idealistic, legislative and pragmatic) for primary (a) and secondary (b) motivation of coral reef restoration projects.** Number of projects: n = 12 for primary and n = 11 for secondary motivation.

**Table 3. Summary of total annual project costs, spatial extent of coral reef area intervened, project duration and feasibility from 12 case studies in the Spanish-speaking Caribbean and Eastern Tropical Pacific (Fundación Grupo Puntacana's restoration programs were treated as two independent projects).**

|  | Total cost per year (2018 USD) | Spatial extent (ha) | Project duration (yrs) | Feasibility (best guess) |
|---|---|---|---|---|
| Median | 93,000 (± 32,731) | 1.00 (± 1.30) | 3.0 (± 1.5) | 0.7 (± 0.03) |
| Min | 10,000 | 0.06 | 1 | 0.5 |
| Max | 331,802 | 8.39 | 17 | 0.8 |
| N | 11 | 7 | 12 | 11 |

Error is given as standard error (± SE). Abbreviation: number of observations (N).

which are already being implemented or are in the initiation phase. These projects were identified through an open call for participation at the Reef Futures conference in December 2018, which aimed to bring together a large international community to develop and implement solutions to the global coral reef crisis.

We describe 12 coral reef restoration case studies in the Caribbean and Eastern Tropical Pacific that employ coral reef restoration techniques including direct transplantation, coral gardening, micro-fragmentation and larval propagation (S1 Table in S1 File). With a median total project cost per year of $93,000 USD, spatial extent of 1 ha, duration of 3 years and overall project feasibility of 0.7, we show that coral reef restoration projects in these countries are less expensive than previously thought, have transitioned from small- to large scale of restoration intervention, have persisted for a long time and have achieved higher success rates compared to values from systematic reviews on this topic [19, 21, 51]. For instance, the most recent published literature review on coral reef restoration presented a median value of $400,000 (2010 USD) to restore 1 ha (10,000 m$^2$) of coral reef, project duration of 1 year, an area intervened of 0.01 ha (108 m$^2$), and survival of restored corals as an item-based success indicator of 0.61 [19].

Although recommended by the best practices for ecological restoration by the Society for Ecological Restoration [5], not many studies in the published literature report on specific and measurable indicators to track success and progress of the restoration. Here we report on biotic and socio-economic indicators such as the number of coral transplants grown and outplanted, increase in cover or density, the number of local dive shops engaged and the number of fishermen trained in maintaining the nurseries and monitoring the outplanting sites. The variety of success indicators reported here have a time-component and go beyond survival as the only metric for assessing the overall restoration progress which was criticized by the published literature as a metric for overall project feasibility [51].

The objectives for coral reef restoration are often undocumented in the published literature, thus extracting data on the objectives from published papers may lead to skewed results. For example, Hein et al. [50] reviewed 83 published coral reef restoration studies and observed that 60% of the studies reported on evaluating the biological response of the coral reef ecosystem to transplantation (outplanting) as a main project objective. The remaining 40% of studies included the following objectives: 1) to accelerate reef recovery post-disturbance (18%), 2) to re-establish a self-sustaining, functioning reef ecosystem (48%), 3) to mitigate coral loss prior to a known disturbance (18%), and 4) to reduce population declines and ecosystem degradation (15%). In comparison, we observed that when data are elicited directly from restoration practitioners, most coral reef restoration projects in the Caribbean and Eastern Tropical Pacific had the following objectives: 1) to optimize or scale-up restoration approaches (51.1%), followed by 2) to provide alternative, sustainable livelihood opportunities (14.9%). Similarly, the projects presented here were mostly motivated by biotic reasons such as to enhance biodiversity and experimental reasons (both 41.7%), followed by idealistic/pragmatic reasons (both 8.3%). In contrast, most motivations to restore coral reefs extracted from the published literature were dominated by experimental reasons, such as to improve the restoration approach and answer ecological research questions (65.3%) [19]. Unlike terrestrial restoration which has been practiced for centuries and is grounded in restoration ecology, the restoration of coral reef restoration is relatively new and originates from experimental biology. Hence it may not be surprising that the experimental rationale is still one of the predominant ones reported from published studies [19]. For the studies reported here, other motivations (e.g. biotic, pragmatic or idealistic) take over once the project aims at operationalising and scaling-up a functional coral reef restoration approach. Many restoration projects presented here focused on harnessing social or economic benefits from coral reef restoration such as involving the

community through inclusion in activities or educational programs to raise awareness or to provide alternative, sustainable livelihood opportunities for local communities. An assessment of social, economic, and cultural benefits derived from the restoration of coral reefs has been largely ignored by the published literature, which has mostly concentrated on outcomes related to the ecology or described endeavours to improve restoration technology [19]. The present work is an attempt to bridge the gap between academics and practitioners. Academics tend to be more focused on small-experimental coral reef restoration attempts to answer questions of ecological concern, whereas practitioners are more focused on optimising and scaling-up restoration. Bridging the gap between academics and practitioners has been identified as critical for many fields of conservation [54, 55].

Coral reef restoration in the Caribbean and Eastern Tropical Pacific face challenges similar to those of restoration efforts elsewhere in the world. For instance, the Intergovernmental Panel on Climate Change (IPCC) concluded that, if no action is taken to reduce $CO_2$ emissions, coral reefs would decline by 70–90% with global warming of 1.5°C above pre-industrial levels, whereas virtually all coral reefs ($>$ 99 percent) would be lost with 2°C warming within the next 50 years [56]. Thus, while actions to reduce $CO_2$ emissions are drastically needed, restoration with more heat tolerant species is regarded as a key strategy to rehabilitate the ecological function and ecosystem services provided by coral reefs [46]. In addition to climate change, coral reef restoration in the Caribbean and ETP face other challenges such as overfishing, sedimentation, pollution, and non-sustainable coastal development [57–62]. The recent outbreak of Scleractinian Coral Tissue Loss Disease (SCTLD) has decimated coral populations and is of major concern to those attempting to restore corals in the Caribbean. Since its onset in 2017, SCTLD has caused widespread mortality of corals, especially in the Florida Reef Tract and the Gulf of Mexico [63, 64]. The vectors causing this disease or how it can be prevented are currently unknown but are most likely bacterial [63]. A further challenge to the restoration of coral reefs in the Caribbean and ETP is the apparent lack of funding and funding strategies. None of the countries have cohesive national plans for the restoration of coral reefs similar to the Reef Restoration and Adaptation Plan in Australia which has invested AUD $100 million in 2018 to develop, trial, and deploy coral reef restoration interventions for the Great Barrier Reef (GBR) [65].

Despite the impediment of limited financial resources, considerable advances in coral reef restoration, both scaling-up of interventions and optimisation of techniques, have been achieved in Colombia, Costa Rica, Dominican Republic, Mexico and Puerto Rico. Identifying all major players from the coral reef restoration consortium in the Caribbean and ETP and how connected the restored reef sites are will be essential for understanding the recovery of degraded coral communities. For instance, one of the largest and longest running projects (18 years) has plans to restore up to 8.4 ha, requiring outplanting 10,000 corals or up to 8,000 coral settlement bases with coral larvae per year. These interventions were led by pioneering environmental NGOs and foundations, who often procured un-paid volunteers to carry out much of the work. The interventions were also enabled by strong partnerships initiated by the champion organization with universities (e.g. Universidad Nacional Autónoma de México, Centro de Investigación y de Estudios Avanzados del Instituto Politécnico Nacional, University of Puerto Rico, Universidad del Valle, Universidad Javeriana de Cali, Universidad de Costa Rica), conservation management bodies and regulators (e.g. Natural Parks administrations, Departments of Natural and Environmental Resources and the United States National Oceanic and Atmospheric Administration), associations (e.g. Fishers Association, Caribbean Hotel and Tourism Association), national and international business partners (e.g. Experiences Xcaret), international environmental NGOs (e.g. Conservation International, The Nature Conservancy, SECORE International), tourist service providers (e.g. the Iberostar Group), private

donations (e.g. Global Giving), international grant schemes (e.g. from Deutsche Gesellschaft für Internationale Zusammenarbeit, Counterpart International, InterAmerican Development Bank (IDB)) and in large part with local community groups. Coral reef restoration still remains an underfunded area in the Spanish-speaking countries and territories of the Caribbean and ETP despite the ecosystem services restored coral reefs could provide for the regions such as food, tourism income, protection against storms and wave surges [66, 67], and reduction in insurance premiums by offering coastal protection [68].

There are a few caveats that need to be considered when assessing the data within the present work. First, this review does not contain an exhaustive list of interventions in the Spanish-speaking countries and territories of the Caribbean and ETP. Additional projects exist or are planned, but were not aware of, or chose to not participate in our open call. Second, the projects presented here varied in their specific objectives, best practice protocols, and monitoring, which hindered their comparison. For example, some projects were designed to improve and optimise the restoration approach (experimental projects), while others were more operational, i.e., aimed to scale-up the restoration of coral reefs by using already established restoration techniques. Furthermore, the projects used different best practice protocols or key indicators of restoration success, such as size of transplant and density of transplants which made a direct comparison between the projects difficult. Some projects lacked monitoring milestones to evaluate the survival, cover and health conditions of outplanted corals beyond year one. Yet, post-restoration monitoring is an imperative method needed to confirm that outplanted corals are self-sustaining which, from an evolutionary perspective, is the ultimate goal of any restoration effort [5–7]. Third, evaluation of the overall project feasibility or the likelihood of success to reach specific project objectives is naturally linked to local conditions and circumstances, thus may be a subjective measure directly related to the experience of the practitioner. More quantitative measures of overall project feasibility (e.g., based on measurements) would be a considerable improvement over the qualitative (derived from expert elicitation) approach.

Prior to any conservation action, a prioritisation of interventions based on decision-support frameworks is recommended to help practitioners increase their planning rigor, project accountability, stakeholder participation, transparency in decisions, and learning [69]. Cost-effectiveness analysis is such a tool that allows for the evaluation and prioritisation of conservation interventions [70]. This analysis relates the costs of a project to its key outcomes or benefits i.e., the specific measures of project effectiveness [70, 71]. Although this work includes all data required for a cost-effectiveness analysis (see Supplementary material), we considered that comparing the different projects against each other will be inappropriate given the variety of their project objectives (e.g. experimental vs. operational) and the lack of standardisation in reporting on cost, feasibility and key outcomes.

Future collaborations between academics, local communities and practitioners will be crucial if we want to achieve restoration at meaningful ecological, spatial and social scales [72]. Unfortunately, the language barrier often inhibits such collaborations. For instance, Amano et al. [73] argues that languages are still a major barrier to global science by showing that more than 35% of the knowledge in conservation is missed by those who only look at peer-reviewed literature in English. Many practitioners who carry out large-scale coral restoration projects only convey their knowledge in the form of unpublished reports and grey literature [19], which adds another level of complexity to the loss of information on restoration efforts. Here we close this gap by accessing this knowledge and overcoming the language barrier.

## Conclusions

Although not previously highlighted by the published literature, there are many coral reef restoration projects currently in progress in the Spanish-speaking countries and territories of the Caribbean and Eastern Tropical Pacific. Most of these projects are being carried out by pioneering civil organizations often in strong partnerships with universities, conservation management bodies and regulators, tourism operators, the private sector, associations, and local community groups. While coral reef restoration has been portrayed as too expensive and challenging with regards to spatial scale, duration, and success, the projects presented here have shown that many of these barriers have already been overcome. These pioneering endeavours were often possible by in-kind commitments of staff and volunteers as well as involvement of the local community and tourism operators, thus socio-economic aspects play a substantial role in coral reef restoration in the Caribbean and Eastern Tropical Pacific. Strong national plans for restoration in conjunction with national and international funding are needed to multiply the already existing activities made by Latin-American organisations to improve the health and status of coral reefs in the Caribbean and Eastern Tropical Pacific. From this compilation of data and knowledge, it is apparent that it would be beneficial for coral reef restoration practitioners in this area to coordinate their efforts with each other and make sure they are sharing and implementing their best practices protocols to standardise efforts and track restoration progress by specific, measurable, achievable and repeatable metrics of success through time.

## Supporting information

**S1 File.**
(DOCX)

**S1 Data.**
(XLSX)

## Acknowledgments

We would like to thank Nufar Charuvi, the pioneer driving the Alianza Coralina Taganga project, who, although no longer with us, continues to inspire by his example, dedication, and passion he served over the last decade for Colombian coral reefs and the local community. This manuscript has been developed upon in-kind time of the authors and has not received any financial support. We acknowledge the Iberostar Group for covering the open access publication fees of this manuscript.

## Author Contributions

**Conceptualization:** Elisa Bayraktarov, Anastazia T. Banaszak, Sarah Frías-Torres.

**Data curation:** Anastazia T. Banaszak, Phanor Montoya Maya, Joanie Kleypas, Jesús E. Arias-González, Macarena Blanco, Johanna Calle-Triviño, Nufar Charuvi, Camilo Cortés-Useche, Victor Galván, Miguel A. García Salgado, Mariana Gnecco, Sergio D. Guendulain-García, Edwin A. Hernández Delgado, José A. Marín Moraga, María Fernanda Maya, Sandra Mendoza Quiroz, Samantha Mercado Cervantes, Megan Morikawa, Gabriela Nava, Valeria Pizarro, Rita I. Sellares-Blasco, Samuel E. Suleimán Ramos, Tatiana Villalobos Cubero, María F. Villalpando, Sarah Frías-Torres.

**Formal analysis:** Elisa Bayraktarov.

**Investigation:** Elisa Bayraktarov, Anastazia T. Banaszak, Phanor Montoya Maya, Joanie Kleypas, Jesús E. Arias-González, Macarena Blanco, Johanna Calle-Triviño, Nufar Charuvi, Camilo Cortés-Useche, Victor Galván, Miguel A. García Salgado, Mariana Gnecco, Sergio D. Guendulain-García, Edwin A. Hernández Delgado, José A. Marín Moraga, María Fernanda Maya, Sandra Mendoza Quiroz, Samantha Mercado Cervantes, Megan Morikawa, Gabriela Nava, Valeria Pizarro, Rita I. Sellares-Blasco, Samuel E. Suleimán Ramos, Tatiana Villalobos Cubero, María F. Villalpando, Sarah Frías-Torres.

**Methodology:** Elisa Bayraktarov, Anastazia T. Banaszak, Phanor Montoya Maya, Joanie Kleypas, Jesús E. Arias-González, Macarena Blanco, Johanna Calle-Triviño, Nufar Charuvi, Camilo Cortés-Useche, Victor Galván, Miguel A. García Salgado, Mariana Gnecco, Sergio D. Guendulain-García, Edwin A. Hernández Delgado, José A. Marín Moraga, María Fernanda Maya, Sandra Mendoza Quiroz, Samantha Mercado Cervantes, Megan Morikawa, Gabriela Nava, Valeria Pizarro, Rita I. Sellares-Blasco, Samuel E. Suleimán Ramos, Tatiana Villalobos Cubero, María F. Villalpando, Sarah Frías-Torres.

**Project administration:** Elisa Bayraktarov.

**Resources:** Elisa Bayraktarov.

**Software:** Elisa Bayraktarov.

**Supervision:** Elisa Bayraktarov, Sarah Frías-Torres.

**Visualization:** Elisa Bayraktarov.

**Writing – original draft:** Elisa Bayraktarov, Anastazia T. Banaszak, Phanor Montoya Maya, Joanie Kleypas, Macarena Blanco, Johanna Calle-Triviño, Nufar Charuvi, Miguel A. García Salgado, Edwin A. Hernández Delgado, Samantha Mercado Cervantes, Megan Morikawa, Gabriela Nava, Valeria Pizarro, Tatiana Villalobos Cubero, Sarah Frías-Torres.

**Writing – review & editing:** Elisa Bayraktarov, Anastazia T. Banaszak, Phanor Montoya Maya, Joanie Kleypas, Jesús E. Arias-González, Macarena Blanco, Johanna Calle-Triviño, Camilo Cortés-Useche, Victor Galván, Miguel A. García Salgado, Mariana Gnecco, Sergio D. Guendulain-García, Edwin A. Hernández Delgado, José A. Marín Moraga, María Fernanda Maya, Sandra Mendoza Quiroz, Samantha Mercado Cervantes, Megan Morikawa, Gabriela Nava, Valeria Pizarro, Rita I. Sellares-Blasco, Samuel E. Suleimán Ramos, Tatiana Villalobos Cubero, María F. Villalpando, Sarah Frías-Torres.

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
