## [Decision Letter · Decision Letter 0]

15 Apr 2020

PONE-D-20-01163

Coral reef restoration efforts in Latin American countries and territories

PLOS ONE

Dear Dr Bayraktarov,

Thank you for submitting your manuscript to PLOS ONE. After careful consideration, we feel that it has merit but does not fully meet PLOS ONE’s publication criteria as it currently stands. Therefore, we invite you to submit a revised version of the manuscript that addresses the points raised during the review process.

Dear Authors

As you can see, both the reviewers have recommended major revision and i concur with them. Please go through the comments and suggestions carefully and revise manuscript accordingly. If you do submit the revised manuscript, I will read it decide whether it goes out for 2nd round or not 

We would appreciate receiving your revised manuscript by May 30 2020 11:59PM. To enhance the reproducibility of your results, we recommend that if applicable you deposit your laboratory protocols in protocols.io, where a protocol can be assigned its own identifier (DOI) such that it can be cited independently in the future. For instructions see: http://journals.plos.org/plosone/s/submission-guidelines#loc-laboratory-protocols

We look forward to receiving your revised manuscript.

Kind regards,

Shashank Keshavmurthy, PhD

Academic Editor

PLOS ONE

Journal Requirements:

1. Please include captions for your Supporting Information files at the end of your manuscript, and update any in-text citations to match accordingly. Please see our Supporting Information guidelines for more information: http://journals.plos.org/plosone/s/supporting-information

2. Thank you for icnluding your competing interests statement; "The authors have declared that no competing interests exist."

We note that one or more of the authors are employed by a commercial company: “SECORE International, Inc., Iberostar Hotels & Resorts”

Additional Editor Comments (if provided):

Dear Authors,

you can see that 2 reviews have submitted their comments and suggestions and both of them think that the manuscript can be a good modification provided, it is revised considerably. Please go through them and revise your manuscript accordingly and then I will get it back to both of them or one of them for the 2nd round

Reviewers' comments:

Reviewer's Responses to Questions

**Comments to the Author**

1. Is the manuscript technically sound, and do the data support the conclusions?

Reviewer #1: Partly

Reviewer #2: Yes

2. Has the statistical analysis been performed appropriately and rigorously? 

Reviewer #1: I Don't Know

Reviewer #2: N/A

3. Have the authors made all data underlying the findings in their manuscript fully available?

Reviewer #1: No

Reviewer #2: Yes

4. Is the manuscript presented in an intelligible fashion and written in standard English?

Reviewer #1: Yes

Reviewer #2: Yes

5. Review Comments to the Author

Reviewer #1: The authors presented data from 12 coral reef restoration case studies from five Latin American countries, and described their motivations and techniques used, and provided estimates on total annual project cost per unit area of reef intervened, spatial extent as well as project duration. They also presented the relative success of the restoration efforts based on the perspective of the researchers. This study is important as relatively few have reported data on restoration success in this region. However, to complete the paper and make it acceptable for publication, authors should also discuss the success of their efforts based on the survival and growth of transplants. Please see attachment for additional comments.

Reviewer #2: The paper by Bayraktarov and co-authors presents unpublished data from 12 coral reef restoration case studies from five Latin American countries. The authors describe the motivations and techniques used, and provide estimates on project duration, total annual project cost, spatial extent and likelihood of success. This paper is timely and provides some much needed synthesis of ongoing coral reef restoration efforts from Latin America, that might otherwise be confined to grey literature or not published in a way that is widely accessible. Without these syntheses it is very hard to make progress in improving the outcomes of restoration projects. This paper, therefore, makes a valuable contribution to the restoration literature. Generally, I thought the paper was well written and the authors have done a good job of synthesising several very different case studies in a comparable way. I have a few general comments and some specific comments below:

General comments:

1) I feel that the paper could be improved by providing a clearer definition of ecological restoration and rehabilitation and introducing the importance of setting goals in judging restoration success. 2) Data presented in the main table are very useful, but there is not really anything on actual success rates of each project. I realise many of the projects are ongoing, but it would be good when possible to give some idea of how successful some of these projects have been at meeting their stated goals, (e.g., how many outplanted corals have survived, what increases in coral cover have been achieved etc.?). This is crucial as it is that, that will have an effect on actual costs. It might also be good distinguish between project planned costs and actual value for money (i.e., the cost per hectare of reef successfully restored).

Specific comments:

Line 53: “We found that most projects used direct transplantation, the coral gardening method, micro-fragmentation or larval propagation.” Could you put percentages here as you have done in for other categories.

Line 57: “Reasons for restoring coral reefs were mainly biotic and experimental (both 42%), followed by idealistic and pragmatic motivations (both 8%).” Not clear what is meant here? You go on to explain this in the text, but on its own in the abstract it does not make much sense without some explanation. I’m also not convinced that “experimental restoration” should be considered restoration in the strict sense. You really need some clear definitions of restoration early on in the introduction and need to cite the most recent SER guidelines. I think you need to clearly distinguish between restoration ecology and ecological restoration and define what you mean by a restoration project.

Lines 68-69: “The goal of any restoration action is to eventually establish self‐sustaining, sexually reproducing populations with enough genetic variation enabling them to adapt to a changing environment [3-5].” I think the authors should expand a little bit here. In the latest SER principles and standards (Gann et al 2019 in Restoration Ecology), they go into quite a lot of detail about what constitutes restoration and rehabilitation. There’s also quite a bit of debate about whether restoration should be defined strictly as restoring an ecosystem to a match that of a reference native ecosystem, or whether restorative activities (e.g., rehabilitation) should also be considered under the broad definition of restoration (e.g., see response to Higgs et al 2019). Here you boil it down to one definition, but that may not be the goal of all reef restoration efforts. I think the key point here is that any restoration effort should have a clear goal, but what the precise goals are should be determined by the practitioners rather than being pre-prescribed.

Line 75: “Management programmes have not aided in the recovery of A. palmata [9].”

I feel like you really should say a little more here to justify this statement. You need a sentence or two here about why traditional management failed to restore Acropora in the Atlantic? You also need to say here why people think that restoration will succeed where other management initiatives have failed? In many cases, are conditions really suitable for restoration? I think the rationale is that, where conditions are suitable to support populations of Acropora (i.e., water quality, herbivory etc. are adequate) but larval supply/or post-settlement survival are inadequate, seding new populations to kick start recovery may actually be feasible.

Line 91: “In situ nurseries are typically located at well-lit sites..”

Why should they be well lit? Please explain this rationale.

Line 94: “Eat biofouling?”

Not sure if this is best wording. Remove biofouling through grazing perhaps?

Lines 118-120: “While efforts in the USA, Australia or places where European scientists conduct their research are well described in the published literature and disseminated at conferences, there is a paucity of documentation on reef restoration projects carried out by practitioners in the Caribbean and Eastern Tropical Pacific.”

This is true, but actually there are still many poorly documented efforts in Asia, so a similar synthesis is needed there. Furthermore, in Australia, restoration has only just begun and so not much data available.

Line 108: “which are settled onto substrates and then transported and seeded onto a degraded coral reef [31-33].”

Can I suggest also citing Guest et al 2014 (Coral Reefs volume 33, pages45–55) here as it was one of first studies to design a specific settlement substrate, use it to settle and grown corals in nurseries and on the reef and to follow these corals through until maturity. It also provides contrast the Chamberland paper which tests a non-attached substrate method versus the Guest et al paper, which is an attached substrate method.

Line 112: “Without the need of laboratory facilities [34]”.

That is only if you do all of the culturing in situ, you would still need lab facilities if you did the initial larval rearing ex situ. For balance, you could cite Edwards et al 2015. (Direct seeding of mass-cultured coral larvae is not an effective option for reef rehabilitation. Mar Ecol Prog Ser 525:105-116. https://doi.org/10.3354/meps11171) here, as this was the first paper to attempt larval seeding and follow through with later monitoring. Perhaps you could also cite the original study that attempted larval seeding on the reef, e.g., Enhancement of coral recruitment by in situ mass culture of coral larvae A. J. Heyward, L. D. Smith, M. Rees, S. N. Field, MEPS 230:113-118 (2002) doi:10.3354/meps230113.

Line 115: “Also, they do not cause damage to the parent colonies.”

This is only true if gametes are collected in situ with nets or from spawn slicks. If colonies are removed from the reef, as is the case in many published studies, they often do not survive being outplanted back to the reef subsequently.

Line 148: “The motivations for each restoration project were adopted from [10, 37, 38] and classified as biotic, experimental, idealistic, legislative, and pragmatic (Table 1).”

You need to make clear here distinction between experimental (restoration ecology) and actual (ecological restoration). They are different things and the costs, scale and reasons for doing restoration ecology are completely different from ecological restoration.

Line 243: “we show that coral reef restoration projects in these countries are more cost

effective, have overcome the barriers of scaling-up restoration interventions, are persistent through time, and have a higher likelihood of success than reported from previous literature [10, 12, 40].”

This sentence not very clear. What are you saying here?

Line 280: “restoration with heat resilient species.” I think you need to define what is meant by heat resilient here. I’m also not sure if this is the best terminology as resilience has a specific meaning in ecology. Can I suggest “more heat tolerant species” as an alternative?

6. PLOS authors have the option to publish the peer review history of their article (what does this mean?). If published, this will include your full peer review and any attached files.

Reviewer #1: No

Reviewer #2: Yes: James Guest

---

## [Author Response · Author response to Decision Letter 0]

9 Jun 2020

Response to Reviewers on “Coral reef restoration efforts in Latin American countries and territories” submitted to PLOS ONE Collection on Biodiversity Conservation

***The Reviewers’ comments are in bold while the authors’ responses are in plain text. Line numbers refer to track changes version of the manuscript. ***

Reviewer #1

Reviewer #1: The authors presented data from 12 coral reef restoration case studies from five Latin American countries, and described their motivations and techniques used, and provided estimates on total annual project cost per unit area of reef intervened, spatial extent as well as project duration. They also presented the relative success of the restoration efforts based on the perspective of the researchers. This study is important as relatively few have reported data on restoration success in this region. However, to complete the paper and make it acceptable for publication, authors should also discuss the success of their efforts based on the survival and growth of transplants. Please see attachment for additional comments.

Response: We acknowledge the positive feedback of the Reviewer and their time. We have reworked the manuscript based on their suggestions and think that it is much stronger in its revised version. We have added a new column to Table 2 where we outline success indicators for all restoration projects and elaborate on these in the discussion. Those indicators of success are very different in nature because some projects are operational while some other are experimental and not all projects are based on transplantation of fragments. They are informed by the objectives.

This was possible for all projects except for the Alianza Coralina Taganga because the person driving this work has sadly passed away recently.

Line 41: Insert "coral" before "bleaching."

Response: We have changed bleaching to coral bleaching. See Line 39.

Line 41: recovery of what?

Response: We have changed this text to: “where the natural recovery of an ecosystem is negligible”. See Line 39

Table 2: This table is very useful, but it is not easy to read. Instead of inputting whole paragraphs per column, authors can come up with concise phrases to convey the important information in each paragraph. For instance, under Colombia in the techniques employed, the authors may just mention floating mid-water nursery. Then, the longer description can be summarized and submitted as supplementary material.

Response: Initially, we had dedicated a section for each restoration project, however the editor urged us to add these sections into the supplementary material. We have now substantially shortened the paragraphs per column and added any descriptive text to supplementary material.

Line 205: These values (percentages) would be easy to follow if presented in a graph.

Response: We have now added a graph illustrating the percentages of the different motivations’ categories of the restoration projects. We have done this for the primary and secondary motivations. Please see Figure 3.

Line 293: The current consortium is also useful in identifying how connected their reef sites are - which may be useful in understanding recovery of degraded coral communities.

Response: The reviewer highlights a very important point here. We have added a discussion of connectivity in Lines 353 – 355 of the revised manuscript:

“Identifying all major players from the coral reef restoration consortium in the Caribbean and ETP and how connected the restored reef sites are will be essential for understanding the recovery of degraded coral communities.”

Line 560: What the authors are presenting are relative success of the restoration efforts based perspective. It would be best to also present data on survival of coral transplants and cover, especially that these groups are transplanting locally vulnerable coral species.

Response: This comment was also made by Reviewer 2. We have now added a new column in Table 2 to quantify the relative success of the restoration efforts of each project and elaborate on this point in the discussion. See Lines 265 – 273:

“Although recommended by the best practices for ecological restoration by the Society for Ecological Restoration [5], not many studies in the published literature report on specific and measurable indicators to track progress of the restoration (Bayraktarov et al. In review. MERS). Here we report on biotic and socio-economic indicators such the number of coral transplants grown and outplanted, increase in cover or density, the number of local dive shops engaged and the number of fishermen trained in maintaining the nurseries and monitoring the outplanting sites. The variety of success indicators reported here have a time-component and go beyond survival as the only metric for assessing the overall restoration progress which was criticized by the published literature as a metric for overall project feasibility [50].”

Reviewer #2: The paper by Bayraktarov and co-authors presents unpublished data from 12 coral reef restoration case studies from five Latin American countries. The authors describe the motivations and techniques used, and provide estimates on project duration, total annual project cost, spatial extent and likelihood of success. This paper is timely and provides some much needed synthesis of ongoing coral reef restoration efforts from Latin America, that might otherwise be confined to grey literature or not published in a way that is widely accessible. Without these syntheses it is very hard to make progress in improving the outcomes of restoration projects. This paper, therefore, makes a valuable contribution to the restoration literature. Generally, I thought the paper was well written and the authors have done a good job of synthesising several very different case studies in a comparable way. I have a few general comments and some specific comments below:

Response: We thank the Reviewer for their positive feedback and valuable comments that helped us improve the revised version of this manuscript significantly.

General comments:

1) I feel that the paper could be improved by providing a clearer definition of ecological restoration and rehabilitation and introducing the importance of setting goals in judging restoration success. 2) Data presented in the main table are very useful, but there is not really anything on actual success rates of each project. I realise many of the projects are ongoing, but it would be good when possible to give some idea of how successful some of these projects have been at meeting their stated goals, (e.g., how many outplanted corals have survived, what increases in coral cover have been achieved etc.?). This is crucial as it is that, that will have an effect on actual costs. It might also be good distinguish between project planned costs and actual value for money (i.e., the cost per hectare of reef successfully restored).

Response: We have now added a definition of the term ‘ecological restoration’ in the introduction for which we followed the SER Primer and their International Standards for the Practice of Ecological Restoration. We have also provided a definition of the term ‘rehabilitation’ and have highlighted the importance of goal setting in ecological restoration. The revised section in Ln 67 – 72 now reads:

“Active restoration is defined as the process of assisting the recovery of an ecosystem that has been degraded, damaged, or destroyed [1]. It may be increasingly necessary on coral reefs, once it has been determined that the natural recovery of corals is hindered [2]. In comparison, rehabilitation is typically described as the replacement of structural or functional characteristics of an ecosystem that have been diminished or lost [3]. As for any conservation intervention, setting clear goals and defining indicators to measure progress towards these goals is of pivotal role in judging success [4].”

We have now added a new column in Table 2 where we describe the multiple ‘Indicators of Success’ of the restoration projects. Those indicators of success are very different in nature because some projects are operational while some other are experimental and not all projects are based on transplantation of fragments. They are informed by the objectives.

Reporting on total cost for a restoration is very difficult and several papers including de Groot et al. 2013 have criticised that cost values from the published literature are disparate and are never compiled and reported in a comprehensive and standardised way. The project leads attempted to follow the best practices by Iacona et al. 2018 when they carried out estimation of the total cost for their projects; however, comparing those between the projects might not be possible due to distinct project objectives. The authors consider that distinguishing between project planned costs and actual value for money (i.e., the cost per hectare of reef successfully restored) would require expert advice from an Economist specialised in the valuation of conservation interventions and ecosystem services – this clearly goes beyond the scope of the present manuscript, which is to highlight coral reef restoration efforts in Latin America that are unknown to the international scientific community.

Specific comments:

Line 53: “We found that most projects used direct transplantation, the coral gardening method, micro-fragmentation or larval propagation.” Could you put percentages here as you have done in for other categories.

Response: Thank you for this suggestion. We have now added the percentages of techniques applied by the restoration projects. The revised section in Ln 236 – 240 now reads:

“The restoration projects use techniques that include direct transplantation (one project, 9%), coral gardening (7 projects, 64%), micro-fragmentation (5 projects, 45%), and larval propagation (2 projects, 18%) (Figure 2). Some projects also apply a combination of techniques e.g. direct transplantation, coral gardening and micro-fragmentation or coral gardening and micro-fragmentation as well as coral gardening and larval propagation (Supplementary information Table S1)”

Line 57: “Reasons for restoring coral reefs were mainly biotic and experimental (both 42%), followed by idealistic and pragmatic motivations (both 8%).” Not clear what is meant here? You go on to explain this in the text, but on its own in the abstract it does not make much sense without some explanation. I’m also not convinced that “experimental restoration” should be considered restoration in the strict sense. You really need some clear definitions of restoration early on in the introduction and need to cite the most recent SER guidelines. I think you need to clearly distinguish between restoration ecology and ecological restoration and define what you mean by a restoration project.

Response: Thank you for pointing this out. We agree with the reviewer and have now defined the terminology around active restoration vs rehabilitation by referring to the SER guideline documents. See LN 67 – 74, the opening lines of the introduction.

Lines 68-69: “The goal of any restoration action is to eventually establish self‐sustaining, sexually reproducing populations with enough genetic variation enabling them to adapt to a changing environment [3-5].” I think the authors should expand a little bit here. In the latest SER principles and standards (Gann et al 2019 in Restoration Ecology), they go into quite a lot of detail about what constitutes restoration and rehabilitation. There’s also quite a bit of debate about whether restoration should be defined strictly as restoring an ecosystem to a match that of a reference native ecosystem, or whether restorative activities (e.g., rehabilitation) should also be considered under the broad definition of restoration (e.g., see response to Higgs et al 2019). Here you boil it down to one definition, but that may not be the goal of all reef restoration efforts. I think the key point here is that any restoration effort should have a clear goal, but what the precise goals are should be determined by the practitioners rather than being pre-prescribed.

Response: Thank you for pointing this out. We have also addressed this point in our response to Reviewer 1. We have added definitions on ecological restoration and referred to why it is important to clearly state the goals of restoration and implement indicators to report on success. See LN 67 – 72 early on in the introduction.

Line 75: “Management programmes have not aided in the recovery of A. palmata [9].”

I feel like you really should say a little more here to justify this statement. You need a sentence or two here about why traditional management failed to restore Acropora in the Atlantic? You also need to say here why people think that restoration will succeed where other management initiatives have failed? In many cases, are conditions really suitable for restoration? I think the rationale is that, where conditions are suitable to support populations of Acropora (i.e., water quality, herbivory etc. are adequate) but larval supply/or post-settlement survival are inadequate, seeding new populations to kick start recovery may actually be feasible.

Response: Thank you for your insightful comment and suggestions. We have replaced this statement with the following (see lines 81-89): 

“The lack of natural recovery of Caribbean coral reefs [11] has spurred the need for active management programmes to assist in their recovery [12, 13]. Management actions include effective spatial planning, enforcement, no take zones, treatment of sewage and protection of adjoining ecosystems such as mangroves [12, 14, 15]. Resilience based management of coral reefs [16] may stimulate coral recovery, especially if applied in conjunction with active restoration [13, 17]. The rationale being that seeding corals onto reefs where larval supply or post-settlement survival have been inadequate, will only be successful if the conditions are suitable for supporting their survival and growth.”

References:

Andersson AJ, AA. Venn, L Pendleton, A Brathwaite, EF. Camp, S Cooley, D Gledhill, M Koch, S Maliki, C Manfrino. 2019 Ecological and socioeconomic strategies to sustain Caribbean coral reefs in a high-CO2 world. Regional Studies in Marine Science, 29:100677 doi.org/10.1016/j.rsma.2019.100677.

Mcleod E, K.R.N. Anthony, P.J. Mumby, J. Maynard, R. Beeden, N.A.J. Graham, S.F. Heron, O. Hoegh-Guldberg, S. Jupiter, P. MacGowan, S. Mangubhai, N. Marshall, P.A. Marshall, T.R. McClanahan, K. Mcleod, M. Nyström, D. Obura, B. Parker, H.P. Possingham, R.V. Salm, J. Tamelander 2019. The future of resilience-based management in coral reef ecosystems. Journal of Environmental Management 233: 291-301

Rinkevich B. 2005 Conservation of Coral Reefs through Active Restoration Measures: Recent Approaches and Last Decade Progress Environ. Sci. Technol. 2005, 39, 12, 4333–4342

Roff G, Mumby PJ. 2012 Global disparity in the resilience of coral reefs. Trends Ecol Evol. 27(7):404-13. doi: 10.1016/j.tree.2012.04.007

Steneck RS., Arnold SN., Boenish R, de León R, Mumby PJ., Rasher DB., Wilson MW. 2019 Managing Recovery Resilience in Coral Reefs Against Climate-Induced Bleaching and Hurricanes: A 15 Year Case Study From Bonaire, Dutch Caribbean. Frontiers in Marine Science 6:265 

Walsworth, T.E., Schindler, D.E., Colton, M.A. et al. Management for network diversity speeds evolutionary adaptation to climate change. Nat. Clim. Chang. 9, 632–636 (2019). https://doi.org/10.1038/s41558-019-0518

Line 91: “In situ nurseries are typically located at well-lit sites..”

Why should they be well lit? Please explain this rationale.

Authors: We have changed this section in LN 115 – 117 to clarify:

“In situ nurseries are typically located in sheltered environments where conditions are favourable for coral growth and safe from predation, storm surges, and wave energy, and are regularly maintained and cleaned by physical removal of algal growth [22].”

Line 94: “Eat biofouling?”

Not sure if this is best wording. Remove biofouling through grazing perhaps?

Authors: We have changed this in LN 117 – 119 to:

“However, strategic siting of ocean nurseries can promote the recruitment of fish assemblages that remove biofouling through grazing, thus may significantly reduce person-hours spent in nursery cleaning [23].”

Lines 118-120: “While efforts in the USA, Australia or places where European scientists conduct their research are well described in the published literature and disseminated at conferences, there is a paucity of documentation on reef restoration projects carried out by practitioners in the Caribbean and Eastern Tropical Pacific.”

This is true, but actually there are still many poorly documented efforts in Asia, so a similar synthesis is needed there. Furthermore, in Australia, restoration has only just begun and so not much data available.

Response: We completely agree with the reviewer, however, since this manuscript focuses on Latin-American countries and because we (the authors of this paper) do not have enough knowledge about what practitioners are doing in Asia or Australia (if they don’t publish) to make such a broad-reaching comment, we prefer to allow others with much more in-depth knowledge about the coral reef restoration efforts in these regions to discuss a potential data paucity there.

Line 108: “which are settled onto substrates and then transported and seeded onto a degraded coral reef [31-33].”

Can I suggest also citing Guest et al 2014 (Coral Reefs volume 33, pages45–55) here as it was one of first studies to design a specific settlement substrate, use it to settle and grown corals in nurseries and on the reef and to follow these corals through until maturity. It also provides contrast the Chamberland paper which tests a non-attached substrate method versus the Guest et al paper, which is an attached substrate method.

Response: We are grateful to the Reviewer for providing a key reference that we have missed. We have included this reference. See LN 142.

Line 112: “Without the need of laboratory facilities [34]”.

That is only if you do all of the culturing in situ, you would still need lab facilities if you did the initial larval rearing ex situ. For balance, you could cite Edwards et al 2015. (Direct seeding of mass-cultured coral larvae is not an effective option for reef rehabilitation. Mar Ecol Prog Ser 525:105-116. https://doi.org/10.3354/meps11171) here, as this was the first paper to attempt larval seeding and follow through with later monitoring. Perhaps you could also cite the original study that attempted larval seeding on the reef, e.g., Enhancement of coral recruitment by in situ mass culture of coral larvae A. J. Heyward, L. D. Smith, M. Rees, S. N. Field, MEPS 230:113-118 (2002) doi:10.3354/meps230113.

Response: We have added this section to synthesize the literature suggested by the reviewer, now in LN 140 – 142 of the revised version. Note that we have also added the dela Cruz and Harrison 2017 publication for balance.

“The first attempts to use larval seeding on the reef have been developed only recently (in 2002, [38]) and it is still a matter of active debate whether direct seeding of mass-cultured coral larvae is an effective option for reef rehabilitation [39, 40].”

Line 115: “Also, they do not cause damage to the parent colonies.”

This is only true if gametes are collected in situ with nets or from spawn slicks. If colonies are removed from the reef, as is the case in many published studies, they often do not survive being outplanted back to the reef subsequently.

Response: The reviewer highlights an important point here – we have added in LN 146 – 147:

“Also, they do not cause damage to the parent colonies when gametes are collected in situ with nets or from spawn slicks without removing the gravid colonies from their location”

Line 148: “The motivations for each restoration project were adopted from [10, 37, 38] and classified as biotic, experimental, idealistic, legislative, and pragmatic (Table 1).”

You need to make clear here distinction between experimental (restoration ecology) and actual (ecological restoration). They are different things and the costs, scale and reasons for doing restoration ecology are completely different from ecological restoration.

Response: This is an excellent point, which we have tried to highlight in LN 314-320:

“Unlike terrestrial restoration which has been practiced for centuries and is grounded in restoration ecology, the restoration of coral reef restoration is relatively new and originates from experimental biology. Hence it may not be surprising that the experimental rationale is still one of the predominant ones reported from published studies [12]. For the studies reported here, other motivations (e.g. biotic, pragmatic or idealistic) take over once the project aims at operationalising and scaling-up a functional coral reef restoration approach.”

Line 243: “we show that coral reef restoration projects in these countries are more cost

effective, have overcome the barriers of scaling-up restoration interventions, are persistent through time, and have a higher likelihood of success than reported from previous literature [10, 12, 40].”

This sentence not very clear. What are you saying here?

Response: We have changed this sentence to:

“We show that coral reef restoration projects in these countries are less expensive than previously thought, have transitioned from small- to large scale of restoration intervention, have persisted for a long time and have achieved higher success rates compared to values from systematic reviews on this topic [10, 12, 40].” See LN 285-288.

Line 280: “restoration with heat resilient species.” I think you need to define what is meant by heat resilient here. I’m also not sure if this is the best terminology as resilience has a specific meaning in ecology. Can I suggest “more heat tolerant species” as an alternative?

Response: We appreciate the suggestion made by the reviewer and have replaced “restoration with heat resilient species” with “more heat tolerant species” in this sentence. See LN 337.

---

## [Editor Report · Decision Letter 1]

22 Jun 2020

Coral reef restoration efforts in Latin American countries and territories

PONE-D-20-01163R1

Dear Dr. Bayraktarov,

We’re pleased to inform you that your manuscript has been judged scientifically suitable for publication and will be formally accepted for publication once it meets all outstanding technical requirements.

Kind regards,

Shashank Keshavmurthy, PhD

Academic Editor

PLOS ONE
---

## [Editor Report · Acceptance letter]

7 Jul 2020

PONE-D-20-01163R1 

Coral reef restoration efforts in Latin American countries and territories 

Dear Dr. Bayraktarov:

I'm pleased to inform you that your manuscript has been deemed suitable for publication in PLOS ONE. Congratulations! Your manuscript is now with our production department. 

Kind regards, 

on behalf of

Dr. Shashank Keshavmurthy 

Academic Editor

PLOS ONE